# Metabolic Pathway Analysis of Nitrogen and Phosphorus Uptake by the Consortium between *C. vulgaris* and *P. aeruginosa*

**DOI:** 10.3390/ijms20081978

**Published:** 2019-04-23

**Authors:** A. Suggey Guerra-Renteria, M. Alberto García-Ramírez, César Gómez-Hermosillo, Abril Gómez-Guzmán, Yolanda González-García, Orfil González-Reynoso

**Affiliations:** 1Chemical Engineering Department, Centro Universitario de Ciencias Exactas e Ingenierías, Universidad de Guadalajara, Blvd. M. García Barragán # 1451, C.P. 44430 Guadalajara, Jalisco, Mexico; suggey_89@hotmail.com (A.S.G.-R.); Cesar.GomezH@cucei.udg.mx (C.G.-H.); agomezguzm@gmail.com (A.G.-G.); 2Electronics Department, Centro Universitario de Ciencias Exactas e Ingenierías, Universidad de Guadalajara, Blvd. M. García Barragán #1421, C.P. 44430 Guadalajara, Jalisco, Mexico; seario@gmail.com; 3Wood, Pulp and Paper Department, Universidad de Guadalajara, Km. 15.5 Carretera Guadalajara-Nogales, Las Aguilas, C.P. 45010 Zapopan, Jalisco, Mexico; yolacea@yahoo.com

**Keywords:** extreme pathways, nutrient removal, *C. vulgaris*, *P. aeruginosa*

## Abstract

Anthropogenic activities have increased the amount of urban wastewater discharged into natural aquatic reservoirs containing a high amount of nutrients such as phosphorus (Pi and PO${}_{4}^{-3}$), nitrogen (NH${}_{3}$ and NO${}_{3}^{-}$) and organic contaminants. Most of the urban wastewater in Mexico do not receive any treatment to remove nutrients. Several studies have reported that an alternative to reduce those contaminants is using consortiums of microalgae and endogenous bacteria. In this research, a genome-scale biochemical reaction network is reconstructed for the co-culture between the microalga *Chlorella vulgaris* and the bacterium *Pseudomonas aeruginosa*. Metabolic Pathway Analysis (MPA), is applied to understand the metabolic capabilities of the co-culture and to elucidate the best conditions in removing nutrients. Theoretical yields for phosphorus removal under photoheterotrophic conditions are calculated, determining their values as 0.042 mmol of PO${}_{4}^{-3}$ per g DW of *C. vulgaris*, 19.43 mmol of phosphorus (Pi) per g DW of *C. vulgaris* and 4.90 mmol of phosphorus (Pi) per g DW of *P. aeruginosa*. Similarly, according to the genome-scale biochemical reaction network the theoretical yields for nitrogen removal are 10.3 mmol of NH${}_{3}$ per g DW of *P. aeruginosa* and 7.19 mmol of NO${}_{3}^{-}$ per g DW of *C. vulgaris*. Thus, this research proves the metabolic capacity of these microorganisms in removing nutrients and their theoretical yields are calculated.

## 1. Introduction

Diverse human activities have increased the amount of urban wastewater effluents discharged in natural aquatic reservoirs, containing a high amount of nutrients such as NH${}_{4}^{+}$, NO${}_{3}^{-}$ and PO${}_{4}^{3-}$ as well as organic contaminants. These compounds have been identified as the main cause of eutrophication in natural aquatic reservoirs. Microalgae alone or in co-culture with bacteria offer a promising approach to removing and to re-using nutrients such as nitrogen and phosphorus, since they are possible to be assimilated into the biomass [1]. The phenotypic potential of these co-cultures is essential to establish the theoretical nutrient uptake. It has been reported elsewhere that the microalga *Chlorella* accumulates nitrogen and phosphorus to reach concentrations ranging between 5.0 to 10.1% and 0.5 to 1.3%, respectively [1]. The advantages of using microalgae for this purpose includes the low-cost growing process by considering solar energy. The microalgae metabolic capability can use carbon and nutrients present in wastewater and assimilate them; algae can later be used as a fertilizer or compost avoiding a sludge handling problem [2]. In addition to the wastewater effluent treatments, microalgae can be used for biodiesel production and even considered as a food source, thus, making the tertiary wastewater treatment a sustainable and affordable process [3,4,5,6].

Nevertheless, pure microalgae cultures are only possibly maintained in the lab. In environmental conditions, microalgae always coexist with endogenous bacteria [7]. Hence, the spontaneous interactions between those microorganisms is a regular occurence. In one hand, bacteria is benefited from the microalgae exudates such as oxygen and starch. On the other hand, the microalgae growth is promoted by bacterial products such as carbon dioxide (CO${}_{2}$), inorganic substances and some growth factors [8,9]. However, the practical use of interactions between selected algae and bacteria species in wastewater treatment might be considered to be a technology. There are a few experimental reports that have been working on different consortia of microalga species and bacteria for urban and industrial wastewater treatments [2,10]. The *C. vulgaris* and *P. putida* consortium inherently has shown to be a good nutrient and organic contaminants removal option in synthetic municipal wastewater compared to axenic cultures [11,12]. Lananan, et al. 2014, reported a removal efficiency up to 99.15% of the total phosphorus concentration in domestic wastewater treatment by using *Chlorella* with an effective microorganism (EM-1) co-culture [13]. Moreover, it also has been reported that co-culture of *C. vulgaris* with *Azospirillum brasilense* in cellular immobilization increases the removal of ammonia and phosphorus [8]. Co-culture of *Chlorella* with other bacteria removed up to 80% of total nitrogen present in animal feed wastewater production. However, it has been reported in experimental studies that pure cultures have no effect whatsoever in to removing either nitrogen or phosphorus in industrial wastewater [14]. These studies have demonstrated that the microalgae-bacteria consortium is a milestone biological system to remove nutrients rather than pure cultures of those microorganisms studied under aseptic laboratory conditions. *Pseudomonas* is a common bacteria present in wastewater mentioned in a broad variety of studies [15,16]. However, the metabolic activity and capability of such microorganisms can be altered by varying the culture conditions in the wastewater processes by including those associated with microflora, particularly α-proteobacteria group, such as *Pseudomonas*. To the knowledge of the authors, there is a lack of studies in regard to the interaction between *C. vulgaris & P. aeruginosa*, that are focused on the metabolisms, the upper and lower bounds for nutrients removal according to the biochemical network and the possible metabolic phenotypes.

Currently, the genomic information for specific microorganisms is available from biological databases that are collected from high-throughput technologies. Those databases describe the metabolisms and a broad variety of components such as genes, proteins and other metabolites. Therefore, genome-scale biochemical reaction networks for microorganisms are possible to reconstruct and analyze using metabolic engineering tools [17]. Varma et al. (1993) [18], were the first authors to model a metabolic network for an entire organism (*E. coli*), using Flux Balance Analysis (FBA) to obtain the optimal carbon-flux distribution. Another technique used to analyze genome-scale metabolic networks is the Metabolic Pathway Analysis (MPA), it focuses in finding the phenotypic capabilities by numerically analyzing a set of systematically independent and unique Extreme Pathways (ExPas) [19]. The ExPas are mathematically derived vectors that are used to characterize the phenotypic potential of a defined metabolic network [19,20]. ExPas describe the substrates conversion into products while creating all byproducts required to maintain the systemic elemental balance and the cofactor pools at steady state [17,19]. By calculating ExPas from a metabolic network, it is possible to explain the active metabolisms in a particular pathway and the theoretical yield with respect to carbon sources or nutrients. Thus, by calculating and analyzing ExPas from the metabolic network of the consortium between *C. vulgaris* and *P. aeruginosa*, it is possible to estimate the phenotypic potential under different schemes. Our research group has been working in the evaluation of nutrient removal by using different microorganisms and the consortium between them [12]. This paper presents a metabolic approach to improving our understanding on biological systems where microalgae and bacteria coexist as it usually occurs in most natural waters, which could eventually be implemented in tertiary wastewater treatment.

## 2. Results and Discussion

### 2.1. Stoichiometric Matrix S

All the metabolites, internal and exchange fluxes such as photons (Pho), external glucose (GLUext), sulfate (SO${}_{4}$ext), magnesium (Mg), potassium (K), iron (Fe), calcium (Ca), zinc (Zn), copper (Cu), manganese (Mn) and the key nutrients that are studied in this article (PO${}_{4}$ext, Piext NO${}_{3}$ext and NH${}_{3}$ext) are featured in the 286 × 293 stoichiometric matrix *S*. The output fluxes are: biomass from each microorganism, polyhydroxyalkanoates, maltose, carbon dioxide (CO${}_{2}$ext) and oxygen production (O${}_{2}$ext).

Figure 1 shows the matrix *S* obtained from the reconstructed metabolic model at genomic scale of the particular microalgae-bacteria consortium. The abscissa axis represents both the internal and exchange fluxes and the ordinate axis denotes the metabolites in order of appearance in the stoichiometric model. Moreover, it also represents the metabolism for each microorganisms as shown in Figure 1. Hence, it clearly depicts that each external nutrient is strongly related to a particular biochemical flux, and how the nutrients are incorporated into the metabolisms for each microorganisms.

### 2.2. Extreme Pathways Analysis Featuring the Uptake of Phosphorous Species

Chemical species of phosphorus are presented in two forms: phosphorus (Piext) and phosphate (PO${}_{4}$ext). The phosphorus (red circle in Figure 1, (•)) is required by both microorganisms. First, it could enter as an external flux into endogenous phosphorus (Pi${}_{-}$Cv) as part of the requirements for photosynthesis metabolism in the microalga; subsequently, Pi${}_{-}$Cv takes part on other 52 biochemical reactions that are strongly related to glycolysis, Krebs cycle and oxidative phosphorylation. Therefore, Pi${}_{-}$Cv is one of the metabolites that depicts a broad connectivity between fluxes in the matrix *S*, because it is required by microalga as part of the anabolism as shown in Figure 1. For instance, the flux number 2, in red circle (Piext→Pi${}_{-}$Cv), is active in 2844 (100%) of the ExPas obtained. 2572 ExPas are linked to an assimilation by microalga towards biomass generation and 273 are related to maltose production. The 273 ExPas mentioned above can elucidate a commensalism interaction observed in experimental studies by Guo & Tong (2013) [21], where they found a concentration of 90 mg per liter of extracellular organic carbon (TOC) in a microalga and *Pseudomonas* culture, compared to 59 mg per liter obtained in a single algal culture. The TOC’s measurements in the above research are related to the active maltose production in the 273 ExPas, where the metabolic machinery of microalga works for the bacterium by supplying an endogenous carbon source under a photoautotrophic scheme.

The set of 2572 ExPas were analyzed to consider the removal of phosphorus by assimilation into biomass. The theoretical yields for phosphorus uptake were calculated as a function of biomass of each microorganism. The highest yield obtained from over 2572 analized yields was 180.23 mmol Pi${}_{-}$Cv per g DW of *C. vulgaris*. The major uptake corresponds to an ExPA of pure *C. vulgaris* culture within a photoheterotrophic scheme. It requires 50.79 mmol of glucose per g DW of *C. vulgaris* and 1252.06 mmol of photons per g DW of *C. vulgaris* to be held.

For the *P. aeruginosa* metabolism (reactions 148 to 273, see Appendix A), the Piext (•) enters as part of the oxidative phosphorylation of ATP synthesis. Simultaneously, it is incorporated in 28 biochemical reactions that feature the metabolisms of glycolysis, Krebs cycle and synthesis of acetic acid (Figure 1). In 33% of the total calculated ExPas the maximum theoretical phosphorous uptake by the bacteria was 4.90 mmol Pi${}_{-}$Pa per g DW of *P. aeruginosa*. The maximum yield by the bacteria can occur either in a photoautotrophic or photoheterotrophic scheme of the consortium, the yield value is lower compared to the one obtained by the microalgae.

The phosphate specie (PO${}_{4}$ext) (green circle in Figure 1 (•)), is assimilated by *C. vulgaris* (PO${}_{4}$ext→PO${}_{4-}$Cv) as part of the substrate required for biomass synthesis (flux number 147), it implies a uptake of this species in 90.43% of the ExPas. The yield of one g DW of *C. vulgaris* requires 0.04 m mol of PO${}_{4}$ext if there is microalgae biomass production (Figure 2).

### 2.3. Extreme Pathways Analysis Featuring the Uptake of Nitrogen Species

Nitrogen is represented in two nitrogenous species within the metabolic network; nitrate (NO${}_{3}$ext) and ammonia (NH${}_{3}$ext). Nitrate is first converted into endogenous ammonia (NH${}_{3-}$Cv), in order to be used by the microalgae. The previous condition is considered in flux number 75 (•) and is active in 90.43% of the total ExPas (Figure 2) it has a maximum uptake of 7.19 mmol of NO${}_{3}$ per g DW of *C. vulgaris*. NH${}_{3-}$Cv can be assimilated and incorporated into other 24 fluxes inside the metabolisms of *C. vulgaris*. These fluxes go to the synthesis of amino acids (glutamate, glutamine, glycine, proline, arginine, histidine, isoleucine, leucine, methionine, phenylalanine, chorismate and valine), which are part of the protein synthesis ( flux reaction 148) and nucleic acids (fluxes reactions 140 and 141). The removal of external ammonia (NH${}_{3}$ext, (•) by bacteria is represented in Figure 2 by the purple circle (•). It is present int 33.33% of the total ExPas. The best ExPa predicts a maximum uptake of 10.30 mmol of NH${}_{3}$ per g DW of *P. aeruginosa* either under a photoautotrophic or photoheterotrophic scheme.

### 2.4. The Extreme Pathway with the Highest Yield of Nutrient Uptake

By considering only the ExPas that show a simultaneous uptake of the four nutrient species, 864 feasible ExPas have been obtained; 96 are for photoautotrophic and 768 for photoheterotrophic schemes. Nevertheless, with the amount of mentioned ExPas, it is useful to group them for further analysis, therefore another parameter (organic carbon degradation), was considered to reduce the amount of ExPas. In this case, carbon degradation is represented as external glucose (GLUext), in the metabolic model. The introduction of this criterion reduced from 768 to 336 feasible extremes pathways. Thus, the best ExPas for nutrient removal by the co-culture is depicted as a schematic diagram in Figure 3. It accounts for 246 biochemical flux reactions, 84% of the total metabolic network.

The source of organic carbon in Figure 3 is represented by the glucose (GLUext), while the inorganic carbon is an endogenous source from bacterial respiration (CO${}_{2-}$P). This exchange of metabolites could represent a reduction in the operation costs of tertiary wastewater treatment because some of the oxygen or external carbon dioxide needed will be provided by the co-culture. for The flux of external glucose goes for both microorganisms, having a maximum removal of 0.93 mmol of glucose per g DW of *C. vulgaris* and 13.85 mmol of glucose per g DW of *P. aeruginosa*. Even so, most of the organic source is directed towards the bacterium at glycolysis metabolism. This is in agreement with experimental reports where the growth of bacterium is related to glucose uptake at the expense of microalga development [21]. While the inorganic carbon source comes from the internal respiration of bacteria, the total yield 34.07 mmol of CO${}_{2}$ goes to the Calvin cycle in the microalga metabolism to be fixed into the triose glyceraldehyde 3 phosphate (G3P${}_{-}c$). This last metabolite goes for chlorophyll synthesis and it is also incorporated into the five step of glycolysis. In this scheme, G3P${}_{-}$c is not needed as substrate to produce starch or maltose. The G3P${}_{-}$c metabolite is only used as an important energy compound when there is no carbon dioxide or nutrients. This idea can be reinforced because there is a simultaneous synthesis of biomass of *C. vulgaris* and *P. aeruginosa*. Figure 3 also shows that both microorganisms are not competing between them. Therefore, the best obtained yields for phosphorus and nitrogen uptake by the consortium are: For the algae 19.43 mmol of phosphorus per g DW of *C. vulgaris*, 0.04 mmol of phosphate per g DW of *C. vulgaris* and 7.19 mmol of nitrate per g of DW of *C. vulgaris*. For the bacterium the nutrient removals yields were 4.90 mmol of phosphorus per g DW of *P. aeruginosa* and 10.30 mmol of ammonia per g DW of *P. aeruginosa* were obtained. These results indicate a more efficient uptake of inorganic phosphorous from microalga than from bacterium in the consortium, because of the alga requirements during photosynthetic metabolism according to Figure 1, in fact it is important to mention that all the ExPas (2844) no matter the scheme, presented phosphorus removal, additionally, bacterium only exhibited a phosphorus removal in 33.3% of the total ExPas.

## 3. Methods

### 3.1. Reconstruction of the Genome-Scale Biochemical Reaction Network for the Co-Culture of C. vulgaris and P. aeruginosa

The focus of the first step to rebuild a genome-scale metabolic network for the consortium of *C. vulgaris* and *P. aeruginosa* is to assemble the stoichiometric reactions based on the genome information. Different metabolic databases exist featuring the biochemical reactions with specific genes for each microorganism. In this research, the reconstruction of the metabolic networks was made using the databases BRENDA (BRaunschweig ENzyme DAtabase), NCBI (National Center of Biotechnology Information) and MetaCyc & KEGG, (Kyoto Encyclopedia of referenced literature [4,22,23]).

For microalga, the metabolisms considered are those related to the autotrophic and photoheterotrophic schemes such as photosynthesis, chlorophyll synthesis (*Chla* and *Chlb*), Calvin-Benson cycle, starch metabolism, glycolysis/gluconeogenesis and finally, the basic metabolism for biomass formation such as TCA cycle, fatty acids synthesis, triglycerides synthesis, oxidative phosphorylation, pentose phosphate pathway, protein synthesis (18 amino acids), nucleic acids synthesis, carbohydrate synthesis, glycerophospholipids and maintenance. Such metabolisms are represented by the first 147 biochemical reactions (Appendix A: Metabolic Network of the co-culture of *C. vulgaris* and *P. aeruginosa*, which is available online at Appendix A). Biochemical reactions from 148 to 273 represent the metabolisms for *P. aeruginosa*. For this bacterium the metabolisms considered were those related to the central metabolism in a prokaryotic cell such as starch metabolism, glycolysis, TCA cycle, glyoxylate cycle, pentose phosphate pathway, oxidative phosphorylation, amino acid synthesis, nucleic acid synthesis, peptidoglycan synthesis, synthesis of fatty acids and biomass formation. In a similar way, the metabolisms related to acetic acid and polyhydroxyalkanoates synthesis were included in the metabolic network; the transport and exchange fluxes are represented by reaction 274 to 293 in the metabolic network (Appendix A).

In addition to the information provided by the above mentioned databases, the elementary composition of *P. aeruginosa* was obtained using an elemental analyzer (Fisons model 1108) [24]. These results are used as a milestone to establish the biochemical reaction for the biomass production (biochemical reaction 273, Appendix A: Metabolic Network of the co-culture of *C. vulgaris* and *P. aeruginosa*). Metabolites such as CoA, NAD, NADP, FAD, ADP, and H${}_{2}$O, are not included because they are present in the same concentration as their analogous pairs AcCoA, NADH, NADPH, FADH, and ATP [25]. All stoichiometric coefficients have the units of mmol unless specified as grams. The nomenclature for all compounds is given in Appendix A in Supplementary Materials. Nomenclature of Compounds, which is available online at Appendix A.

### 3.2. Extreme Pathway Analysis (ExPas)

Once the genome-scale metabolic network for the consortium has been rebuilt, the biochemical reactions are ordered in a matrix *S* with m×n dimensions, whose rows (*m*) represent the mass balance for each metabolite (*X*), and *n* is the number of internal and exchange fluxes (ν) participating in each mass balance [20,26,27], respectively.
(1)$$\frac{dX}{dt}=S\cdot \nu$$
ExPas are calculated according to the principles established elsewhere [19,20,26]. Solving the steady-sate constraints
(2)S·ν=0,
and satisfying the biochemical thermodynamics of non-negative internal fluxes
(3)ν≥0,
with the appropriate boundaries α and β for the exchange fluxes *b*, as the nutrient fluxes
(4)$$\alpha_{j}\leq b\leq \beta_{j},$$
the ExPas are obtained using an algorithm developed in the platform of MATLAB (The Mathworks, Inc., Natick, MA, USA). From every single ExPas, the theoretical yield of nutrient uptake is calculated according to the produced biomass for each microorganism. A rigorous analysis is carried out to select the ExPas with the highest theoretical yield for nutrient uptake.

## 4. Conclusions

A genome-scale metabolic network was rebuilt for the co-culture of *C. vulgaris* and *P. aeruginosa*. Theoretical yields of nutrient uptake (nitrogen and phosphorus), are calculated with respect to DW of biomass of each microorganisms under a photoheterotrophic scheme, which can be related to a practical tertiary wastewater treatment. These theoretical yields are important because they could help the design of biological systems knowing the theoretical requirements of oxygen and carbon dioxide to achieve the maximum nutrient uptake. It is important to mention that nutrient uptake yields reported correspond to that ExPA where there is biomass formation for both microorganisms, meaning that the uptake of these nutrients was only by assimilating them into biomass, which means an interaction of symbiosis between these microorganisms and not a competition between them. The ExPa results indicate that there is no formation of other byproducts containing nitrogen or phosphorus under this particular scheme, all nutrient uptake is directed toward microalgae and bacterial growth.

## Figures and Tables

**Figure 1 ijms-20-01978-f001:**
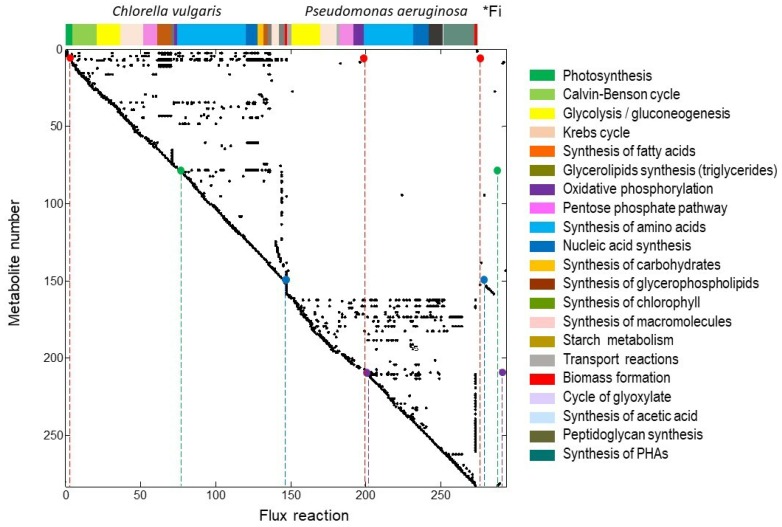
Estequiometric matrix *S*, of co-culture of *C. vulgaris* and *P. aeruginosa*. Main metabolisms of microorganisms are show at the top of the figure; * Fi represents the exchange fluxes considered in the metabolic network -biochemical reactions from 273–293, see Appendix A—Piext is the phosphorous (•), phosphate, PO${}_{4}$ext (•); nitrate, NO${}_{3}$ext (•) and ammonia, NH${}_{3}$ext (•).

**Figure 2 ijms-20-01978-f002:**
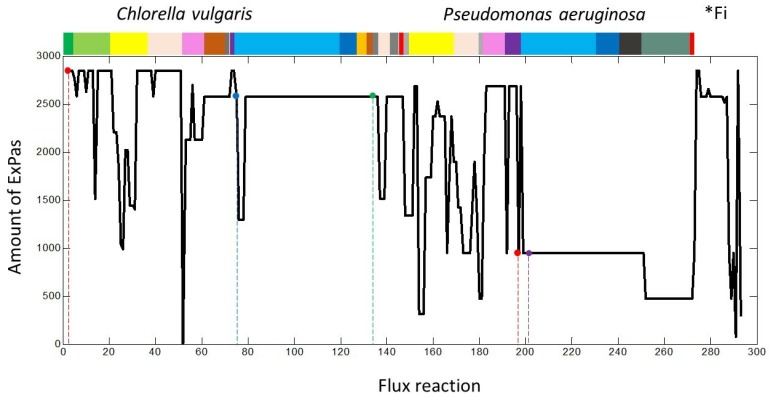
Participation of flux reaction in the extreme pathways. * Fi: exchange fluxes. Piext (•), PO${}_{4}$ext (•), NO${}_{3}$ext (•) and NH${}_{3}$ext (•).

**Figure 3 ijms-20-01978-f003:**
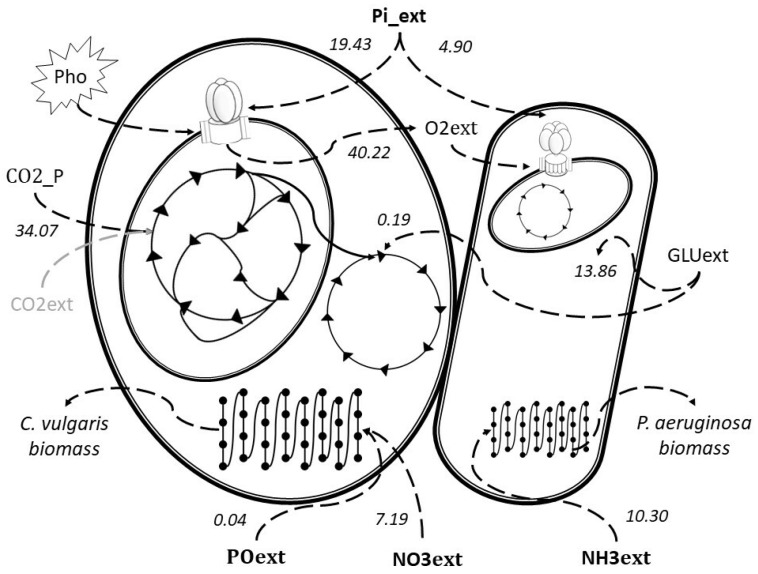
Schematic representation of ExPA with the highest yield of nutrient uptake under photoheterotrophic condition for the consortium of *C. vulgaris* and *P. aeruginosa*. The numbers represent yields of uptaking or producing indicated compounds; the units are mmol of nutrient per g DW of microorganism, respectively; dotted and solid lines represent the exchange and the internal fluxes for each microorganisms.

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
