# Peer review of "Metabolic Pathway Analysis of Nitrogen and Phosphorus Uptake by the Consortium between C. vulgaris and P. aeruginosa"

_ijms, 2019, doi:10.3390/ijms20081978_

Round 1

Reviewer 1 Report

1 Nature of Paper

The paper is based on information from available biological databases and chemical analyses of microbiological metabolism. The purpose and objective of the paper is not cleary described. The paper claims to provide information on “the best conditions in removing nutrients” from wastewater, but provides only theoretical estimates of nitrogen and phosphorus absorption per gram of one algae species and one bacteria species.

2 General Comments

Abstract

There is an insufficiently close connection between the rationale (urban wastewater and pollution of rivers) and the research results. The conclusion that “… this research proves the metabolic capacity of these microorganisms in removing nutrients …” is not new to science. In fact, this has been the basis for wastewater treatment for ages.

Introduction

The initial focus on wastewater treatment is confusing since the paper deals purely with basic microbiology. The introduction chapter should be tailored better to the subject of the paper.

A clear research objective should be stated in the introduction. The current objective (lines 94-96; “fundamental approach to enhance our understanding of biological system”) is too broad.

Results

Most of the information presented in the Results chapter is not reflected in the Conclusions chapter, i.e., weak consistency between results and conclusions.

Discussion (missing)

Discussion of results and comparison to existing knowledge is missing. The only sentence that can be considered to constitute ‘discussion’ is the sentence: “This agree with experimental reports where the growth of a bacterium is related with glucose uptake at the expense of microalgae development [22]” (Lines 178-180 in the Result chapter).

The significance of the core data provided, i.e., the theoretical yields for phosphorus and nitrogen removal by P. aeruginosa and C. vulgaris, should be discussed and compared to similar research. How are these numbers helpful with respect to planning “new strategies for secondary wastewater treatment”? How do these data vary with sunlight and temperature? What is the significance of these data and how can we apply them?

Conclusion

The first paragraph should be moved to the Introduction chapter because it describes justifications for the research, not conclusions.

It appears that the conclusion made from this research is that dissolved N and P are immobilized from wastewater only when both the algae and the bacteria grow and that the extracted N and P were incorporated into the expanding biomass. How will these findings help “finding new strategies for secondary wastewater treatments” as stated in the Introduction chapter?

Appendices

Appendix A and Appendix B are missing.

3 Detailed Comments

Line 19: Replace “… confining in them …” with “… containing… ”.

Lines 25-30: Names of chemical elements should be written in lowercase letters (nitrogen, phophorus).

Line 36: Replace “… confining in them …” with “… containing… ”.

Line 36: Replace “… organics …” with “… organic …”.

Lines 36-37: Replace “… nutrients and organic[s] contaminants 
such as NH4+, NO3− and PO43−“ with 
“… nutrients, such as NH4+, NO3− and PO43−, and organic contaminants“ since the listed ions are inorganic.

Lines 37-38: Replace “… the main cause leading to eutrophication …” with “… the main cause of eutrophication …”. “Cause leading to” is a duplication.

Lines 38-40: Why has “… finding new strategies for secondary wastewater treatments …” received attention? What is wrong with the “old” strategies”. Explain why “new strategies” are needed. The fact that untreated effluents are commonly discarded in natural reservoirs is not a justification. Maybe implementation of old strategies will do the job?

Line 41: Replace “… approach to remove and re-use …” with “… approach to removing and re-using …”.

Line 41-42: ‘Nitrogen’ and phosphorus’ are common nouns and should, therefore, be spelled with lowercase first letters, i.e., ‘nitrogen’ and ‘phosphorus’.

Line 43: Replace “… ranged …” with “… ranging… ”.

Line 46: ‘Carbon’ is also a common noun and should be spelled with a lowercase first letter.

Line 49: Replace “… food source [5]. Thus, making …” with “… food source [5], thus, making …”. The last sentence is not complete and cannot stand alone.

Line 52: The adjective “simultaneous” appears to be redundant in this sentence. Can interactions be non-simultaneous?

Line 66: Replace “… have not effect …” with “… have no effect… ”.

Line 77: Replace “… one specific microorganism …” with “… specific microorganisms …”.

Line 78: Should “…which is collect from …” read “…which is collected from …”?

Line 81: Replace “… authors in modeling …” with “… authors to model …”.

Line 93-94: Write “nutrient removal” instead of “nutrients removal”. A noun used as an adjective should be written in its singular form.

Line 108: Write “abscissa axis” and “ordinate axis” (‘abscissa’ and ‘ordinate’ in sigular form when used as modifiers).

Line 113: Write ‘phosphorus’, not ‘phosphorous’.

Line 114: Regarding the statement “Phosphorous nutrient is presented in two forms; inorganic phosphorous (Piext) and phosphate (PO4ext) …”: Does this suggest that phosphate in solution is not inorganic? What does the term ‘inorganic phosphorus’ include if phosphate is excluded?

Figure 1: “… see Appendix A”: Appendix A is not included in the manuscript.

Figure 3 and Line 177 ++: Use ‘per’ instead of ‘/’ when units are written in the form of text as apposed to symbols, i.e., write “mmol of nutrient per g DW of microorganism”.

Line 198: Replace “… a… networks” with “… a … network”.

Line 229: Appendix B is not shown.

Line 238: Replace “For our knowledge, …” with “To our knowledge, …”.

Lines 249-250: Reconsider the term “elimination of … nutrients”. Nutrients are not eliminated. They are converted from one form to another form and/or transferred from one phase to another phase.

Author Response

We have attached the file in pdf format with the answers to the comments and suggestion to the reviewer 1.  Thank you for your suggestions.

Reviewer 2 Report

Dear Authors!

The manuscript aims to find solutions for cleaning-up wastewaters. As already proposed by other studies, the Authors suggest the use of microalgae with endogenous bacteria. Here the main novelty is reconstruction of genome-scale metabolic network for consortium of these organisms and calculations of the yield. The conclusions are nicely based on the study.

My main, and only comment is about significant improvements in the language of the MS. At first I have started to make exact list of needed corrections, but it would be a really long list. I would suggest asking for a help from a native speaker. Moreover, during improvements, would be good to improve the quality of the Fig 1.

Author Response

(The authors gave the same response as above.)

Round 2

Reviewer 1 Report

General assessment

I regret to conclude that the authors, in my view, have not modified the manuscript to meet the requirements for a scientific publication. My main concern is that the relevance of the study is still obscure with respect to the claimed justification. The answers given by the authors to my question of scientific relevance, do not shed more light on the issue – rather the opposite (see Response 6).

The study appears to be primarily a methodological study since the authors argue that it cannot be compared to any experimental data and since the results do not have any tangible application in wastewater treatment. The authors argue, however, that “this manuscript is important because it is possible the design of operational issues in a tertiary process”. To my understanding, they fail to elaborate on this in the paper.

I recommend, therefore, that the authors present the study in a more relevant context. The paper can, potentially, be published as a methodological study of the determination of nitrogen and phosphorus uptake by the two microbial organisms. This approach would give the paper a more logical framework.

The Introduction is still largely unclear. See detailed comments below. The objective and justification for the study is still primarily explained in the Conclusion chapter, where the information does not belong. The Conclusion chapter should only describe what the authors conclude from the results.

The Results chapter in Version 1 is called Results and Discussion in Version 2. However, we still miss an explanation how the results can help in the design of wastewater treatment. For any practical application of the results, it would be essential to know how P. vulgaris and P. aeruginosa perform compared to other relevant algae/bacteria consortia. What do the presented yield data tell us?

The Conclusion chapter should be more specific. The chapter simply states that the study provides a “reconstruction of a genome-scale metabolic network” and “results of theoretical yields”. In addition, the chapter states that the study shows that nitrogen and phosphorus was removed only by assimilation into biomass. The reader is left with the impression that the authors find the concusion to be elusive.

In addition to my structural concerns, I agree with Reviewer 1’s call for “significant improvements in the language” of Version 1 of the manuscript. Unfortunately, most of the problems remain in Version 2. The weaknesses are not only linguistic but also logical. Consistent with Reviewer 1, who stated that “I have started to make exact list of needed corrections, but it would be a really long list”, the current reviewer has made detailed comments only to the Introduction chapter.

Detailed comments (Introduction only)

Line 35: The sentence “Therefore, it is foreseen novel strategies for tertiary wastewater treatment” is not clear. Maybe better to write “Therefore, novel strategies for tertiary wastewater treatment are needed”?

Line 36: “This kind” is singluar, whereas “strategies” is plural. Write either “these kinds of strategies” or “this kind of strategy”.

Line 36: Regarding the phrase “… received an important attention …”: Only one attention? The general idea of ‘attention’ is probably intended. If so, write “received … attention”. Also, the term ‘important attention’ is awkward. Terms like ‘significant attention’ or ‘extensive attention’ might be better.

Line 40: The phrase: “… accumulates concentrations …” is also awkward. Algae can accumulate nitrogen and phosphorus, but cannot accumulate concentrations. Concentrations increase as nitrogen accumulates.

Line 40: What is “o%”?

Lines 42–44: Awkward sentence: “The microalgae metabolic capability can use endogenous carbon sources as an advantage to recycle the assimilated nutrients to use them as compost by avoiding a sludge handling problem.” Maybe the authors try to express that the microalgae have the metabolic capacity to use endogenous carbon, which may facilitate recycling of assimilated nutrients? Clarity is needed.

Lines 48–49: Unclear logic: “Nevertheless, a pure microalgae culture is not always maintained. Microalgae always coexist with endogenous bacteria”. It appears illogical to state that algae are ‘not always pure’ if they ‘always coexists’ with bacteria. It sounds like algae are never pure and always coexist. Clarification is needed.

Lines 49–50: Another unclear statement: “Hence, it is common to observe spontaneous interactions between those microorganisms.” Does this imply that it is “common to observe spontaneous interactions” because “microalgae always coexist with … bacteria”? Wouldn’t we always observe spontaneous interactions if they always coexist?

Lines 53–54: Awkward statement: “Therefore, natural interactions between microalgae and bacteria could be considered as an innovative technology …“. Natural interactions have existed for millions of years and are, therefore, not “an innovative technology”. However, the practical use of interactions between selected algae and bacteria species in wastewater treatment might be considered to be a technology. In scientific literature, wording is essential. Logical precision is needed.

Line 66: The statement “… rather than pure cultures of those microorganisms” appears to be inconsistent with the earlier statement that “Microalgae always coexist with endogenous bacteria …” (Lines 48–49).

Author Response

Dear, Ms. Josephine Xue
Assistant Editor of International Journal
of Molecular Science.

Again, we would like to thank the reviewer and you for all the time that both of you
have taken in reading our manuscript and for all the help provided in order to make it
better. Thanks to both of you.

It is very important to let you know that we have changed the Title of the manuscript
trying to avoid confounding the readers. The new title is:

Metabolic Pathway Analysis of Nitrogen and Phosphorus Uptake by the Consortium be-
tween C. vulgaris and P. aeruginosa.

We want to let you know that all the suggestion from Reviewer 1 are taking in account
to obtain a better version of our manuscript. In the zip folder, where the latex files are
saved, we include a pdf  fi le showing the main di fferences between the previous and this last
version of the manuscript.

Again, for your comments and suggestions, thank you.

Responses are written in the pdf file.
